# Prenatal Choline Supplementation Alters One Carbon Metabolites in a Rat Model of Periconceptional Alcohol Exposure

**DOI:** 10.3390/nu14091874

**Published:** 2022-04-29

**Authors:** Sarah E. Steane, Vinod Kumar, James S. M. Cuffe, Karen M. Moritz, Lisa K. Akison

**Affiliations:** 1School of Biomedical Sciences, The University of Queensland, St Lucia, QLD 4072, Australia; s.steane@uq.edu.au (S.E.S.); v.kumar1@uq.edu.au (V.K.); j.cuffe1@uq.edu.au (J.S.M.C.); k.moritz1@uq.edu.au (K.M.M.); 2Child Health Research Centre, The University of Queensland, South Brisbane, QLD 4101, Australia

**Keywords:** methyl group, one carbon metabolism, prenatal alcohol, maternal nutrition, placenta, mass spectrometry, DNA methylation

## Abstract

Prenatal alcohol exposure disturbs fetal and placental growth and can alter DNA methylation (DNAm). Supplementation with the methyl donor choline can increase fetal and placental growth and restore DNAm, suggesting converging effects on one-carbon metabolism (1CM). We investigated the impact of periconceptional ethanol (PCE) exposure and prenatal choline supplementation on 1CM in maternal, placental, and fetal compartments. Female Sprague Dawley rats were given a liquid diet containing 12.5% ethanol (PCE) or 0% ethanol (control) for 4 days before and 4 days after conception. Dams were then placed on chow with different concentrations of choline (1.6 g, 2.6 g, or 7.2 g choline/kg chow). Plasma and tissues were collected in late gestation for the analysis of 1CM components by means of mass spectrometry and real-time PCR. PCE reduced placental components of 1CM, particularly those relating to folate metabolism, resulting in a 3–7.5-fold reduction in the ratio of s-adenosylmethionine:s-adenosylhomocysteine (SAM:SAH) (*p* < 0.0001). Choline supplementation increased placental 1CM components and the SAM:SAH ratio (3.5–14.5-fold, *p* < 0.0001). In the maternal and fetal compartments, PCE had little effect, whereas choline increased components of 1CM. This suggests that PCE impairs fetal development via altered placental 1CM, highlighting its role in modulating nutritional inputs to optimize fetal development.

## 1. Introduction

Prenatal alcohol exposure increases the risk of poor pregnancy outcomes, including miscarriage, stillbirth, and fetal growth restriction [1,2,3]. It can also affect the development of the fetal brain, resulting in a condition known as fetal alcohol spectrum disorder (FASD). FASD is characterized by lifelong neurological and physical disabilities and is the most common preventable neurodevelopmental disorder worldwide [4,5,6,7]. Despite health messaging advising against prenatal alcohol use, up to 80% of women consume alcohol during pregnancy, most commonly prior to the recognition of pregnancy [8]. During these first weeks of pregnancy, the preimplantation embryo undergoes epigenetic reprogramming to ‘switch off’ the maternal and paternal genomes via demethylation, followed by the establishment of new methylation marks to activate the embryonic genome [9]. Exposure to alcohol during this critical time can disturb DNA methylation (DNAm) and gene expression in the early embryo, resulting in alterations to the cells that will give rise to the placenta and fetus, affecting their growth and development [10]. Therefore, it has been hypothesized that altered DNAm is one of the mechanisms by which prenatal alcohol exerts its effects and can ultimately contribute to FASD in the offspring [11]. Therapeutic intervention to target these epigenetic modifications is urgently required, with ongoing research investigating choline as a potential candidate.

Choline is a dietary micronutrient with multiple essential roles during fetal development. Choline can modulate DNAm through its interactions with one-carbon metabolism (1CM) which can shift the balance between s-adenosylmethionine (SAM) and S-adenosylhomocysteine (SAH), and therefore alter methylation potential. Choline is also the precursor for membrane phospholipids and the neurotransmitter acetylcholine (Ach) [12]. Prenatal supplementation with choline has been shown to normalize alcohol-induced alterations to offspring DNAm, gene expression, and neurobehavioural outcomes in preclinical and clinical studies [13,14,15,16], as reviewed in [17,18]. However, it has been noted that prenatal choline supplementation in women can increase the production of trimethylamine oxide (TMAO) [19], which has been linked to metabolic syndrome [20] and pre-eclampsia [21].

Using a rat model of periconceptional ethanol (PCE) exposure, we have demonstrated that alcohol exposure around the time of conception alters placental morphology and gene expression and results in fetal growth restriction [22]. Further studies demonstrated that PCE reduced maternal plasma choline at embryonic day (E) 2 and resulted in hypermethylation of the E5 blastocyst cells [23]. Using the same model, we investigated whether maternal choline supplementation could ameliorate the effect of PCE on the late-gestation placenta and fetus [24]. We found that choline supplementation increased fetal growth, particularly in PCE-exposed fetuses, partially ameliorating the PCE-induced growth restriction. In the placenta, choline supplementation increased the region responsible for nutrient exchange (the labyrinth zone), and this was associated with decreased global DNAm and increased expression of the imprinted growth factor insulin-like growth factor 2 (*Igf*2). This suggests that choline deficiency played a role in the epigenetic alterations in the placenta observed in this PCE model, with supplementation increasing placental efficiency and subsequent fetal growth.

Further investigation into the biochemical mechanisms underlying choline supplementation is important in order to understand (a) how it ameliorates the effects of prenatal alcohol exposure on the placenta and fetus, and (b) any unintended consequences of supplementation. Therefore, using the same samples as our previous study, the current study aimed to determine the impact of PCE and choline supplementation on key components of 1CM, as well as the choline derivatives Ach and TMAO in maternal, placental, and fetal compartments (Figure 1).

## 2. Materials and Methods

### 2.1. Animal Treatment and Diets

Ethics approval for all animal experimentation was obtained from the University of Queensland Anatomical Biosciences Animal Ethics Committee (25/03/2015; SBMS/467/14/NHMRC) and complied with the Australian Code for the Care and Use of Animals for Scientific Purposes (2013, 8th Edition). Animal housing conditions, ethanol treatment of dams, and supplementation with choline have been reported elsewhere [22,24]. Briefly, female Sprague Dawley rats were randomly allocated to receive either a control liquid diet (Con, *n* = 28) or a liquid diet containing 12.5% ethanol (*v*/*v*) (PCE, *n* = 29), ad libitum, from one estrous cycle prior to mating until the end of embryonic day (E4) of pregnancy. On E5, the liquid diet was removed and replaced with standard laboratory rat chow (Rat & Mouse Meat-Free Diet, Specialty Feeds, Glen Forrest, WA, Australia) and water ad libitum until the dams were culled at E20. The calculated free choline content of this diet was 1.6 g choline/kg chow, as reported by the supplier, and dams on this diet were designated as the Standard chow groups (Std-Con, *n* = 10 and Std-PCE, *n* = 11). In a subsequent PCE and choline supplementation study [24], PCE and Con dams were placed on a new batch of Rat & Mouse Meat-Free Diet (Specialty Feeds) which was independently tested for choline content and found to be 2.6 g choline/kg chow (via mass spectrometry, Pacific Lab Services, Singapore). A subset of these dams continued on this chow until culling at E20 and were designated Int-Con (*n* = 8) and Int-PCE (*n* = 9). The remaining rats were supplemented from E10 using choline-fortified chow (SF15-043, Specialty Feeds), independently verified to have a concentration of 7.2 g choline/kg chow (by mass spectrometry, Pacific Lab Services). These groups were designated as Supp-Con (*n* = 10) and Supp-PCE (*n* = 9). This resulted in six experimental groups.

### 2.2. Blood and Tissue Collection

Maternal and fetal tissue and blood collection has been previously described [22,24]. Briefly, pregnant dams were anaesthetized at E20 with i.p. 50:50 ketamine:xylazine (0.1 mL/100 g bodyweight) (Lyppard Australia Ltd., Northgate, QLD, Australia). Fetuses and placentas were removed and weighed. Fetal trunk blood was collected from males and females separately and pooled by sex and maternal blood was collected via cardiac puncture. All blood samples were centrifuged at 1700× *g* for 10 min at 4 °C and plasma was removed and stored at −80°C for subsequent analysis. Placental labyrinth and junctional zones were carefully teased apart using forceps. Whole fetal brain, fetal liver, and placental tissues were snap-frozen in liquid nitrogen and stored at −80°C. Fetal sex was confirmed using quantitative polymerase chain reaction (qPCR) as previously described [25]. Although tissues and plasma were collected from the Std chow groups at a different time to the Int and Supp groups, all analyses reported in this study were conducted concurrently and have not been previously reported.

### 2.3. Liquid Chromatography-Tandem Mass Spectrometry (LC-MS/MS)

Components of the 1CM pathway and choline derivatives acetylcholine (Ach) and trimethylamine-oxide (TMAO) were measured using LC-MS/MS. The LC-MS/MS system comprised an API 3200 triple quadrupole mass spectrometer (AB SCIEX, Framingham, MA, USA) coupled with an Agilent 1200 series HPLC system (Agilent Technologies, Mulgrave, VIC, Australia). Compound-dependent and source-dependent mass spectrometer conditions were optimized using an automatic optimization and infusion method for all analytes. For sample preparation, frozen whole fetal brains and 100 mg samples of frozen liver and placental labyrinth were weighed and homogenized in 2 volumes of milliQ water using a FastPrep 5G bead homogenizer (MPBiomedicals, Seven Hills, NSW, Australia). Deuterated d9-choline was added to plasma and tissues homogenates to a final concentration of 1 µg/mL as an internal standard. Samples were de-proteinized via the addition of 3 volumes of ice-cold acetonitrile solution with 2% formic acid and incubated at 4 °C for 15 min. Following centrifugation (14,000 rpm for 10 min at 4 °C), supernatants were transferred to vials for LC-MS/MS analysis. Chromatographic separation of molecules was implemented with an Ascentis Express HILIC column (150 mm × 2.1 mm, 100 Å, 2.7 µm; Supelco, Merck, North Ryde, NSW, Australia) under binary gradient conditions, using mobile phase A (0.1% formic acid in LC grade milliQ water with 10 mM ammonium formate; pH 3) and mobile phase B (0.1% formic acid in acetonitrile) with a 400 µL/min flow rate. Analytes were eluted using the binary gradient: 99% mobile phase B from 0–1 min with linear decrease to 1% from 1 to 6 min and maintained at 1% up until 12 min. Column washing and equilibration were achieved via a linear increase to 99% from 12 to 12.5 min, followed by 99% of mobile phase B up until 15 min. An injection volume of 10 µL was used for method development and sample analysis. Analyses were carried out using electrospray ionization in positive ion-mode with multiple reaction monitoring of the following *m*/*z* transitions: choline 104.2 → 60.1, betaine 118.1 → 58.2, DMG 104.1 → 58.1, folic acid 442.1 → 295.0, THF 446.1 → 299.1, 5MTHF 460.4 → 313.1, methionine 150.0 → 104.1, homocysteine 136.0 → 90.0, SAM 399.2 → 250.0, SAH 385.1 → 87.9, vitamin B2 377.0 → 243.1, vitamin B6 170.0 → 57.1, vitamin B12 678.5 → 147.0, Ach 146.0 → 87.0, and TMAO 75.7 → 58.0. The system control and data acquisition were executed using Analyst software version 1.5.1 (AB SCIEX, Framingham, MA, USA). Peak integration was performed using MultiQuant software version 2.0 (AB SCIEX). The peak area ratio of analyte to internal standard was used for analyses and all peak area ratios were expressed relative to the mean of the male Std-Con group.

### 2.4. Quantitative PCR (qPCR) Analysis

Placental expression of key genes involved in one-carbon metabolism were analyzed via real-time quantitative polymerase chain reaction (qPCR) analysis. The genes examined were methionine synthase (*Mtr*), methionine synthase reductase (*Mtrr*), methylene tetrahydrofolate reductase (*Mthfr*), and betaine-homocysteine methyltransferase (*Bhmt*). The gene encoding the enzyme responsible for the endogenous production of choline, phosphatidylethanolamine methyltransferase (*Pemt*), was also examined. RNA was extracted from ~20 mg of placental labyrinth tissue using RNeasy extraction mini kits (Cat# 74106, Qiagen, Chadstone, VIC, Australia). The purification included an on-column DNA elimination step and RNA was eluted from the column in 30 µL of DNase/RNase-free water. The concentration and purity of RNA was measured using a Nanodrop 2000 spectrophotometer (ThermoFischer Scientific). All RNA samples were diluted to 100 ng/µL and 5 µL (500 ng) was used for cDNA synthesis in a total reaction volume of 10 µL using the iScript Reverse Transcription (RT) Supermix (Bio-Rad Laboratories, Gladesville, NSW, Australia), according to the manufacturer’s instructions. Negative (-RT) reactions were included for all samples. RT reactions were performed on a PCR Express Thermal Cycler (ThermoFisher Scientific). Gene expression was analysed using qPCR on an Applied Biosystems Quantstudio 6 Flex Real-Time PCR System (ThermoFisher Scientific). Each 10 µL reaction contained 10 ng cDNA, QuantiNova SYBR Green PCR Master Mix (Cat #208056, Qiagen) and KiCqStart SYBR Green primers (Sigma-Aldrich Saint Louis, MO, USA). Relative gene expression was determined using the comparative threshold method (∆∆CT) and normalized to the geometric mean of the endogenous controls β-actin (*Actb*) and ribosomal protein L19 (*Rpl19*). Primer details are provided in Appendix A. Fold-change was expressed relative to the average of the Std-Con male group for each assay.

### 2.5. Statistical Analyses

Principal components analysis (PCA) was conducted using data on all components of 1CM to assess the effects of PCE and prenatal choline on the pathway overall using MetaboAnalyst (v5.0) [26]. Prior to analysis, data were log_10_ transformed and mean-centered. Where PCA resulted in the separation of groups, loadings plots were examined to identify the primary drivers of separation. Means were compared between groups for individual 1CM components and gene expression data using GraphPad Prism 8.0 (GraphPad Software, San Diego, CA, USA) and data presented as mean ± SEM. Prior to hypothesis testing, data were tested for normal distributions using the D’Agostino–Pearson test or the Shapiro–Wilk test (for small sample sizes). Where data deviated from a normal distribution for any group, data were transformed, or the appropriate non-parametric test was used. For parametric data, a two-way analysis of variance (ANOVA) was used to analyze the effects of treatment (Con, PCE) and the chow group (Std, Int, Supp). Where significant differences were found between treatment groups or there was a significant interaction, Tukey’s multiple comparisons post hoc test was conducted. For non-parametric data, a Kruskal–Wallis test was used to compare each group, and where overall significant differences were found, Dunn’s multiple comparisons post hoc test was used to identify significantly different groups. The significance level was set at *p* < 0.05 and a trend at *p* < 0.1 for all statistical tests.

## 3. Results

### 3.1. E20 Maternal Plasma: One-Carbon Metabolism Molecules

The PCA plot of 1CM molecules in maternal plasma showed a partial separation of the Int and Supp groups from the Std groups, whereas the Con and PCE-exposed groups were not separated (Figure 2A). This indicates that maternal choline supplementation altered the 1CM pathway, whereas PCE had no major effect. The separation of chow groups was driven primarily by two components of the folate cycle portion of 1CM: 5MTHF and vitamin B2. Prenatal choline supplementation increased the levels of 5MTHF and B2 (Figure 2B,C) as well as other folate cycle components, folic acid and THF, but with no change in B12 (Appendix A). Prenatal choline also increased the choline pathway molecules choline and DMG, but with no change in betaine (Appendix A), whereas components of the methionine cycle, methionine, and vitamin B6, were reduced and the SAM:SAH ratio and Hcy were not altered (Appendix A).

PCE generally had no effect on components of 1CM in maternal plasma. However, there was a significant reduction in 5MTHF and B2 due to PCE, particularly in the Std chow group, as determined by post hoc analysis (Figure 2B,C). Correlation analyses between all molecules of the 1CM pathway demonstrated the strongest correlation to be between 5MTHF and B2 (*r* = 0.82, *p* < 0.0001). Interestingly, the correlation between 5MTHF and choline (*r* = 0.73, *p* < 0.0001) was greater than that between 5MTHF and its precursors, folic acid (*r* = 0.54, *p* < 0.001) and THF (*r* = 0.62, *p* < 0.0001).

### 3.2. Placental Labyrinth: One-Carbon Metabolism Molecules

Both PCE exposure and choline supplementation altered the placental content of all 1CM molecules that were measured. The PCA plot demonstrated a clear separation of all six groups for the placentas of both male (Figure 3A) and female fetuses (Figure 3B), with the Std-PCE group (in green, far left of plot) and the Supp-Con group (in pink, far right of plot) being the most different to one another, as indicated by the greatest separation by principal component 1 (PC1, x-axis). Choline supplementation separated the Int-PCE group from the Std-PCE group, bringing this group closer to the Std-Con group, and additional choline in the Supp-PCE group resulted in a further separation, positioning this group to the right of the Std-Con group. PCA analysis also indicated some subtle sex-specific effects of PCE and the chow group on placenta 1CM, with a greater spread of the data on PC1 in females compared to males, as indicated by the different x-axis scales. Separation of groups was driven primarily by alterations to the SAM:SAH ratio in placentas of both male (Figure 3C) and female fetuses (Figure 3D). PCE reduced the SAM:SAH ratio regardless of the chow group, whereas choline supplementation resulted in an increase, most notably in the control placentas and particularly in males. In the unsupplemented (Std) group, PCE exposure reduced the mean SAM:SAH ratio to 33% of control levels in placentas of male fetuses and 17% in placentas of female fetuses. Maternal choline supplementation increased the SAM:SAH ratio, such that PCE-exposed placentas of both male and female fetuses in the Int and Supp groups had 60% and ~180%, respectively, of the SAM:SAH level seen in the Std Con placentas. The greater impact of PCE on the SAM:SAH ratio in the placentas of female fetuses from the unsupplemented group likely explains the greater spread of the data on the PCA plot.

In terms of methyl donors for the formation of SAM, the greatest impact of PCE was seen on the folate portion of the 1CM pathway. Although maternal choline supplementation increased folic acid in a dose-dependent manner in control placentas, there was a lesser increase in the PCE placentas (Figure 4A,E). A similar effect of PCE was seen for the B12 content of the placentas of male fetuses (Figure 4B), whereas in the placentas of female fetuses, maternal choline supplementation resulted in more uniform increases in B12 (Figure 4F) in both control and PCE-exposed placentas. This pattern of effects was also true for vitamin B2 (Appendix A). 5MTHF was also reduced by PCE exposure in the placentas of both male and female fetuses (Figure 4C,G). However, in the placentas of females, maternal choline supplementation resulted in greater increases in 5MTHF in the PCE-exposed groups compared with controls (Figure 4G). A similar pattern was seen for THF (Appendix A). Choline supplementation restored the levels of choline in PCE-exposed placentas (Figure 4D,H), with similar patterns seen for betaine and DMG (Appendix A).

Consistent with the impact of PCE on placental methyl donors from both the folate and choline portions of 1CM, the methionine cycle was also impacted. Methionine was decreased in the placentas of both male and female fetuses (Appendix A) and Hcy was increased (Appendix A). This suggests reduced remethylation of Hcy to methionine, which is further supported by the reduced SAM:SAH ratio described above. PCE also reduced vitamin B6 (Appendix A), which is necessary for the conversion of Hcy to cysteine. Choline supplementation generally produced the opposite effects to PCE, increasing methionine, vitamin B6, and the SAM:SAH ratio, and reducing Hcy.

### 3.3. Placental Labyrinth: Expression of Genes Encoding One-Carbon Metabolism Pathway Enzymes

Given the widespread changes in 1CM due to PCE and choline in the placenta, the expression of key enzymes of 1CM were examined in the placental labyrinth zone using qPCR (Figure 5). There was generally no effect of PCE on the expression of these genes in the placentas of either male or female fetuses. However, in the placentas of males, there was a significant interaction between PCE and chow group on the expression of *Mtrr*, with PCE exposure resulting in a downregulation in the Std group and an upregulation in the Supp group (Figure 5A). A similar pattern was seen for *Mtr*, although the interaction term did not reach statistical significance (*p* = 0.09, Figure 5B). Both enzymes are necessary for the transfer of a methyl group from 5MTHF to remethylate Hcy to methionine. In the placentas of female fetuses there was no effect of PCE on *Mtrr* or *Mtr* (Figure 5E,F), whereas maternal choline supplementation increased the expression of *Mtrr* and *Mtr* in the placentas of both male and female fetuses. The expression of both *Mthfr* and *Bhmt* were unaffected by PCE in the placentas of either male (Figure 5C,D) or female fetuses (Figure 5G,H). However, maternal choline supplementation increased the expression of *Mthfr* in the placentas of females only, where there was also a trend for increased *Bhmt* expression (Figure 4H, *p* = 0.06). The expression of *Pemt*, the enzyme responsible for endogenous choline production, was not altered by PCE but was increased by choline supplementation in the placentas of females only (Appendix A).

### 3.4. Fetal Plasma: One-Carbon Metabolism Molecules

PCA plots of the 1CM molecules measured in fetal plasma showed the separation of the Std chow groups from the supplemented groups, with no separation of Con and PCE-exposed groups in either males or females (Figure 6A,B). This suggests that choline supplementation altered the 1CM pathway, whereas PCE had little effect. The separation of groups was driven primarily by folic acid from the folate cycle, and the SAM:SAH ratio from the methionine cycle.

Folic acid was increased in both male and female plasma in response to choline supplementation and was reduced by PCE in male plasma only (Appendix A). However, it was difficult to detect in many samples, particularly in the unsupplemented Std group (see notes for Appendix A for more details). For the other components of the folate cycle, B2 was increased in male plasma (Appendix A) and B12 was decreased in female plasma (Appendix A) in response to choline supplementation. THF could not be detected and 5MTHF could only be detected in 27 of 91 fetal plasma samples (11 male and 16 female), precluding any meaningful analysis of the sexes separately. The SAM:SAH ratio was decreased in both male and female plasma (Appendix A) in response to choline supplementation, with no effect of PCE. Other components of the methionine cycle were also altered, with prenatal choline resulting in an increase in methionine in both males and females (Appendix A), an increase in B6 in males only (Appendix A), and an increase in Hcy (Appendix A), although in male plasma this did not quite reach statistical significance (*p* = 0.06). Additionally, prenatal choline supplementation increased all components of the choline pathway in both male and female plasma (Appendix A).

### 3.5. Fetal Liver and Brain: One-Carbon Metabolism Molecules

Analysis of individual components of 1CM showed that prenatal choline generally resulted in decreases in the liver with little effect in the fetal brain, and PCE resulted in few alterations in either tissue (Appendix A). However, folic acid and THF could not be detected in any fetal liver samples and 5MTHF could only be detected in a small number of samples. Similarly, THF, 5MTHF, Hcy, SAM, and SAH could only be detected in a small number of brain samples. Therefore, this precluded any meaningful analysis of the abundance of these molecules in these tissues.

### 3.6. Choline Derivatives in Maternal, Placental, and Fetal Compartments: Acetylcholine and Trimethylamine-Oxide

There was a pronounced effect of both choline supplementation and PCE on Ach in the placentas of both male (Figure 7A) and female fetuses (Figure 7C). Choline supplementation decreased placental Ach in the absence of PCE but had little effect on the Ach content in PCE-exposed placentas. However, in the fetal brain there were no effects of either prenatal choline or PCE on Ach levels (Figure 7B,D). In the maternal plasma, choline supplementation reduced Ach, whereas there was no effect of PCE (Appendix A). In fetal plasma, prenatal choline resulted in an increase in Ach in both males and females (Appendix A), whereas PCE reduced Ach in female plasma only. In the fetal liver, choline supplementation reduced Ach in both males and females (Appendix A), whereas PCE also resulted in an overall reduction in Ach in female fetal livers only.

TMAO was decreased by PCE in the placentas of both male (Figure 8A) and female (Figure 8E) fetuses, regardless of the chow group. Prenatal choline increased TMAO to a greater degree in control placentas than PCE-exposed placentas, particularly in the placentas of males. In the fetal compartment, modest choline supplementation (Int group) did not alter TMAO. However, additional supplementation (Supp group) resulted in a ~15-fold increase in TMAO in both male and female fetal plasma (Figure 8B,F) and a ~7 to 8-fold increase in the fetal liver (Figure 8C,G) and brain (Figure 8D,H). PCE resulted in a reduction in TMAO in the fetal liver of males only. TMAO was not measured in maternal plasma.

## 4. Discussion

Choline has shown promise as a potential therapeutic for preventing or ameliorating the impacts of alcohol exposure during pregnancy. An essential biological process modulated by choline is 1CM, which is a complex biochemical network that integrates cellular nutrient availability with epigenetic control mechanisms [27,28]. This is important during pregnancy, supporting the growth and development of the fetus. 1CM components, including choline and folate, are actively transported across the placenta [29,30] and are metabolically linked as methyl donors for 1CM, with each compensating for deficiency of the other [31,32]. Choline supplementation studies in women have demonstrated high demand for choline during pregnancy [33], and although an important methyl donor, it is preferentially used for phosphatidylcholine synthesis, at the expense of donation to 1CM [34]. Prenatal alcohol consumption is known to alter the methylation of placental and fetal DNA, and one biological process which may link alcohol consumption to perturbed DNA methylation is the disruption of 1CM. Alcohol may impact 1CM by decreasing flux through the folate pathway, which would likely exacerbate the demand for choline-derived methyl groups during pregnancy. Supplementation with additional choline in this case may provide the methyl groups needed to prevent or reduce unwanted changes to DNA methylation. In the current study, we investigated the impact of PCE on 1CM in the maternal, fetal, and placental compartments, and examined whether maternal choline could prevent the PCE-induced changes.

Although we found no major impacts of PCE on 1CM in the maternal plasma, it did result in a persistent reduction in maternal circulating 5MTHF and B2 in late pregnancy, in the unsupplemented group. This likely results from a depletion in maternal liver stores, due to reduced absorption during the eight days of ethanol exposure around conception [35]. Alcohol exposure has been shown to impair folate metabolism at multiple levels. Studies in humans and primates demonstrated reduced folate absorption via downregulation of the reduced folate carrier (RFC) [35], and preclinical studies have demonstrated downregulation of the intestinal B2 transporters RFVT-1 and 2 [36]. Alcohol also reduces the activity of methionine synthase [37]. Choline supplementation increased maternal plasma 5MTHF and B2, normalizing levels in PCE-exposed dams, perhaps by providing an alternate source of methyl groups.

In this study we observed major impacts of early-pregnancy alcohol exposure on the placental content of 1CM components in late gestation. This may be a consequence of the PCE-induced alterations to maternal folates, as adequate maternal folate is essential for early placental development in mice, with the expression of folate transporters and metabolizing enzymes found in trophoblast cells, which later form the placenta [38]. Poor trophoblast invasion and subsequent fetal growth restriction have been demonstrated in mice receiving a folate-deficient diet [39]. We have previously shown that PCE in our rat model resulted in hypermethylation in the nuclei of the early trophoblast cells, and reduced CDX2, a marker of pluripotency [23]. Furthermore, these early impacts of PCE resulted in reduced spiral artery trophoblast giant cells at E13, suggesting defective placental invasion, which is associated with poor establishment of the maternal blood supply and reduced placental nutrient uptake [40]. This is supported by the findings of the current study, demonstrating a marked reduction in the placental content of all 1CM components in late gestation, long after the period of alcohol exposure. Prenatal choline supplementation increased placental components of 1CM, perhaps by reducing the impact of PCE on implantation and invasion and/or by increasing the concentration gradient, facilitating placental uptake. This may be one mechanism through which prenatal choline increased placental efficiency and fetal growth in our previous study [26].

Although PCE reduced 1CM components in the placentas of both males and females, the magnitude of the reduction of some 1CM components differed depending on sex. Our previous studies identified sex differences in the placental responses to PCE [22,23], and sex-specific placental adaptions to in utero perturbations have been well documented in both animal models and clinical studies (reviewed by [41,42]). In this study, our PCA analysis demonstrated greater separation of the six treatment groups in the placentas of females compared with males, indicating that while 1CM was affected more in the placentas of female PCE-exposed fetuses on a background of low choline, supplementation resulted in larger increases in 1CM components in PCE-exposed placentas (versus control placentas), compared with males. While separation was primarily due to alterations in the SAM:SAH ratio, there were subtle sex differences in the effects of choline and PCE on several placental components of 1CM.

In the folate portion of the pathway, choline supplementation normalized the levels of 5MTHF and B12 in PCE-exposed placentas of females to a greater degree than in placentas of males. In the placentas of females, but not males, choline increased the expression of the 5MTHF-generating enzyme, methylenetetrahydrofolate reductase. This may indicate that the placentas of females are more able to adapt to disturbances in maternal folate metabolism, as has been demonstrated previously [43].

In contrast to the folate cycle, components of the choline side of the pathway showed the most pronounced alcohol-induced reductions in the absence of choline supplementation, whereas choline supplementation restored PCE-exposed placentas to the level of the controls. Betaine, the downstream product of choline oxidation, is not thought to contribute methyl groups in the placenta as the catalyzing enzyme, BHMT, is not expressed. However, although the BHMT transcript has not been detected in the human placenta [44], BHMT activity has been measured in the bovine placenta [45]. Interestingly, we found abundant expression of *Bhmt* in the rat placenta, suggesting that betaine-derived methyl groups may indeed be utilized. However, further studies are required to confirm BHMT activity in the rat placenta. Choline increased the expression of PEMT by ~2.5-fold in the placentas of females only, which may suggest the greater investment of SAM into phosphatidylcholine synthesis to support placental and fetal growth and/or increased endogenous choline production to support 1CM.

In terms of the central methionine cycle portion of the 1CM pathway, choline supplementation increased methionine and reduced Hcy, ultimately leading to an increase in the SAM:SAH ratio, which is indicative of increased methylation potential. Our previous study using placentas from the same dams found that this was associated with decreased global DNAm in the placentas of both males and females [24]. This suggests that the large increase in the SAM:SAH ratio was the result of both increased flux through the pathway and decreased transfer of methyl groups to DNA. In addition, this study showed that PCE reduced the SAM:SAH ratio in the placentas of both males and females, whereas our previous study found that PCE resulted in the hypermethylation of DNA in the placentas of males and hypomethylation in the placentas of females, compared with controls, across all chow groups [24]. Clearly the relationship between the SAM:SAH ratio and DNAm is complex, with numerous studies finding both hyper- and hypo-methylation in response to methyl-deficient diets, depending on the tissue (reviewed in [46]). This study indicates that this may also extend to placental sex. Our group and others have found sex-specific alterations in placental DNAm and gene expression following prenatal alcohol exposure [22,23,47], indicating that placental adaptions occur via different pathways in males and females, and may underlie sex differences in offspring health, extending into adulthood [48]. However, ultimately, prenatal choline supplementation increased 1CM components in PCE-exposed placentas to the levels of non-exposed placentas in the unsupplemented group in both males and females. Our previous study, using the same treated animals, displayed a similar pattern of effects on placental efficiency and fetal bodyweight [24].

Despite the pronounced effects of PCE on 1CM in the placentas of both males and females, there were minimal effects of PCE in the fetal compartment and these were restricted to males. Folic acid was reduced in the plasma of male fetuses and there was a small increase in betaine in the brain. However, given that there were no major impacts of PCE on the fetal liver or brain, this suggests that the long-term metabolic and neurobehavioural alterations previously reported in this model of PCE (in the absence of choline supplementation) [49,50,51], may be secondary to placental deficits and subsequent growth restriction. Conversely, choline supplementation resulted in an increase in 1CM components in the fetal plasma, and a decrease in the fetal liver. Given our previous finding of increased fetal bodyweight and liver weight following choline supplementation [24], this suggests that prenatal choline resulted in an increased allocation of resources to promote fetal growth. Further studies are required to determine whether choline supplementation ameliorates the metabolic and neurobehavioural abnormalities observed in adult offspring following PCE exposure.

Aside from its role as a methyl donor, choline is also the precursor for the neurotransmitter Ach, which is important for fetal brain development [52]. Although we found some alterations in Ach in the maternal and fetal compartments, there were no changes to the levels of Ach in the fetal brain. However, there were pronounced effects of PCE and choline on Ach in the placenta. Choline supplementation resulted in a dose-dependent reduction in Ach to ~10% of the levels in unsupplemented control placentas, whereas in PCE-exposed placentas, there was little reduction in Ach across the chow groups. Ach is thought to play a role in placental amino acid transport, cellular proliferation, vasodilation, and parturition [53]. The increased Ach in PCE-exposed placentas may therefore indicate a compensatory mechanism contributing to the increased placental efficiency previously seen in this model [24].

Prenatal choline supplementation in women can increase TMAO in maternal plasma, placenta, and cord blood [19]. Importantly, TMAO has been found to be increased in plasma of women with pre-eclampsia [21] and high levels have been associated with liver and cardiac dysfunction [20]. We found that maternal choline supplementation increased placental TMAO in a dose-dependent manner but was greatly reduced in PCE-exposed placentas compared with controls, perhaps due to alcohol induced alterations in microbiome composition [54]. Modest choline supplementation (Int group) did not increase TMAO in the fetus, but additional supplementation (Supp group) increased TMAO throughout the fetal compartment, highlighting that this level of supplementation may have detrimental impacts.

## 5. Conclusions

Prenatal alcohol exposure around the time of conception resulted in a reduction in all 1CM components in the placenta in late gestation, particularly those in the folate cycle, some of which were also reduced in the maternal and fetal compartments. Maternal choline supplementation increased components of 1CM, restoring levels in PCE-exposed placentas to those of unsupplemented controls. There were also sex-specific effects of both PCE and choline on 1CM that may underlie the alterations to placental DNAm that were previously reported in this model. Choline supplementation reduced Ach in control placentas but not in PCE-exposed placentas, the consequences of which require further investigation. However, importantly, there were no effects on Ach in the fetal brain. This study provides further mechanistic insights into the interactions between prenatal diet and alcohol exposure, and emphasizes the importance of maternal dietary choline intake, particularly when alcohol is consumed around conception.

## Figures and Tables

**Figure 1 nutrients-14-01874-f001:**
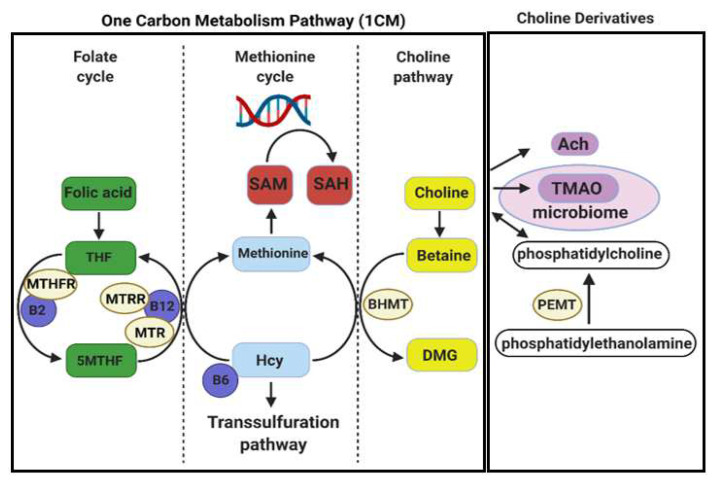
Illustration of the one-carbon metabolism (1CM) pathway and choline derivatives. The 1CM pathway is divided into the folate cycle, methionine cycle, and the choline pathway. The choline derivatives acetylcholine (Ach) and trimethylamine oxide (TMAO), and the endogenous choline synthesis pathway, catalyzed by phosphatidylethanolamine methyltransferase (PEMT), are also shown. THF = tetrahydrofolate; 5MTHF = 5-methyl-tetrahydrofolate; MTHFR = methylenetetrahydrofolate reductase; MTR = methionine synthase; MTRR = methionine synthase reductase; SAM = s-adenosylmethionine; SAH = s-adenosylhomocysteine; Hcy = homocysteine; DMG = dimethylglycine; BHMT = betaine homocysteine methyltransferase.

**Figure 2 nutrients-14-01874-f002:**
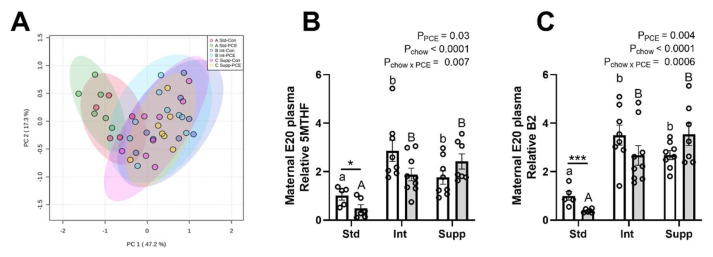
The effects of periconceptional ethanol and choline supplementation on the one-carbon metabolism (1CM) pathway in late gestation maternal plasma. (**A**) Principal components analysis (PCA) of 12 molecules from the 1CM pathway across the 6 experimental groups. Molecules were measured using mass spectrometry. Standard chow (1.6 choline/kg chow, Std) with control (0% EtOH, Con) liquid diet (Std-Con, red); Std with periconceptional ethanol (12.5% *v*/*v* EtOH, PCE) liquid diet (Std-PCE, green); intermediate chow (2.6 g choline/kg chow, Int) with Con (Int-Con, dark blue); Int with PCE (Int-PCE, light blue); supplemented chow (7.2 g choline/kg chow, Supp) with Con (Supp-Con, pink); Supp with PCE (Supp-PCE, yellow). Individual analysis of (**B**) 5-methyltetrahydrofolate (5MTHF) and (**C**) vitamin B2. Open bars are Con and grey bars are PCE (*n* = 5–9 per group). Data expressed as fold-change relative to the Std-Con group (mean ± SEM). Significant differences were identified via two-way ANOVA and Tukey’s post hoc analysis. * *p* < 0.05, *** *p* < 0.001 for differences between Con and PCE within each chow group; chow effects between Con groups are shown by lower-case letters, and between PCE groups by upper-case letters.

**Figure 3 nutrients-14-01874-f003:**
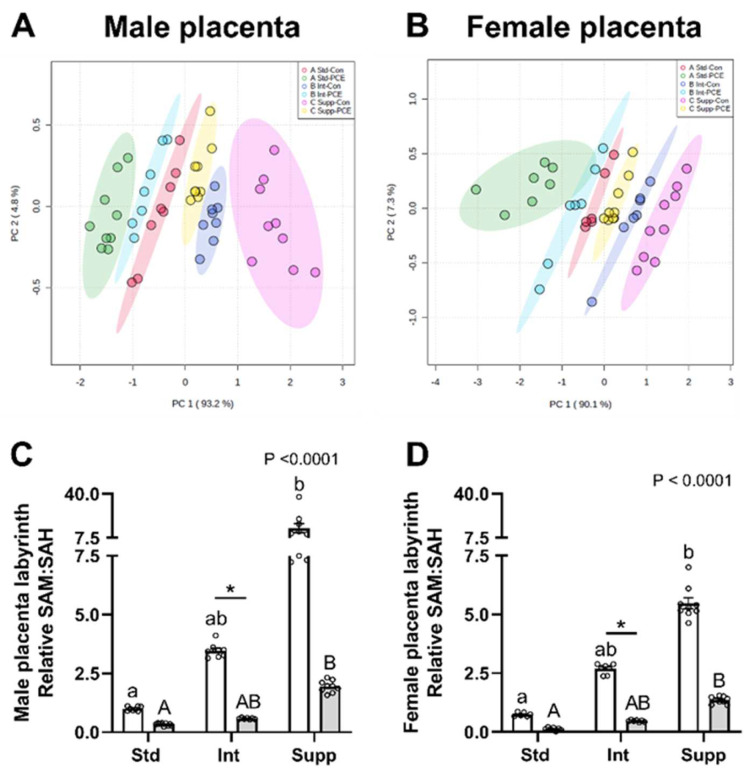
The effects of periconceptional ethanol and choline supplementation on the one-carbon metabolism (1CM) pathway in the placenta. Principal components analysis (PCA) of 12 molecules from the 1CM pathway across the 6 experimental groups in placentas of (**A**) male and (**B**) female fetuses. Molecules were measured using mass spectrometry. Standard chow (1.6 g choline/kg chow, Std) with control (0% EtOH, Con) liquid diet (Std-Con, red); Std with periconceptional ethanol (12.5% *v*/*v* EtOH, PCE) liquid diet (Std-PCE, green); intermediate chow (2.6 g choline/kg chow, Int) with Con (Int-Con, dark blue); Int with PCE (Int-PCE, light blue); supplemented chow (7.2 g choline/kg chow, Supp) with Con (Supp-Con, pink); Supp with PCE (Supp-PCE, yellow). Individual analysis of the s-adenosylmethionine:s-adenosylhomocysteine ratio (SAM:SAH) in placentas of (**C**) male and (**D**) female fetuses. Open bars are Con and grey bars are PCE (*n* = 6–9 per group). Data expressed as fold-change relative to the male Std-Con group (mean ± SEM). Kruskal–Wallis test for non-parametric data was used to identify significant differences across all 6 groups, and Dunn’s post hoc analysis for differences between Con and PCE within each chow group; * *p* < 0.05; differences between Con groups are shown by lower-case letters, and between PCE groups by upper-case letters.

**Figure 4 nutrients-14-01874-f004:**
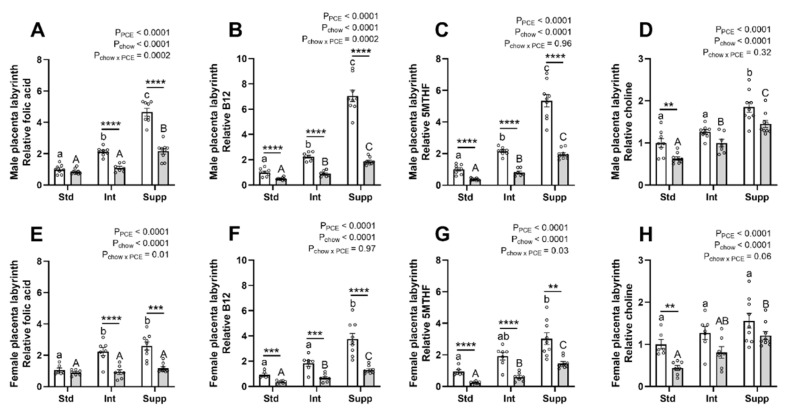
The effects of periconceptional ethanol and maternal choline supplementation on specific components of the folate cycle and choline in placentas of male (**A**–**D**) and female (**E**–**H**) fetuses. (**A**,**E**) folic acid; (**B**,**F**) vitamin B12; (**C**,**G**) 5-methyl tetrahydrofolate (5MTHF); and (**D**,**H**) choline. During the periconceptional period, dams received a control liquid diet (Con; open bars) or a liquid diet containing 12.5% *v*/*v* ethanol (PCE; grey bars). For the remainder of pregnancy, dams received either standard chow (Std), intermediate chow (Int), or supplemented chow (Supp), with increasing levels of choline across these groups (1.6 g, 2.6 g, or 7.2 g choline/kg chow), *n* = 6–9 per group. Molecules were measured using mass spectrometry and data are expressed as fold-change relative to the average of the male Std-Con group (mean ± SEM). Significant differences due to alcohol exposure or chow group were identified by two-way ANOVA and Tukey’s post hoc analysis. ** *p* < 0.01; *** *p* < 0.001, and **** *p* < 0.0001 for differences between Con and PCE within each chow group; chow effects between Con groups are shown by lower-case letters, and between PCE groups by upper-case letters.

**Figure 5 nutrients-14-01874-f005:**
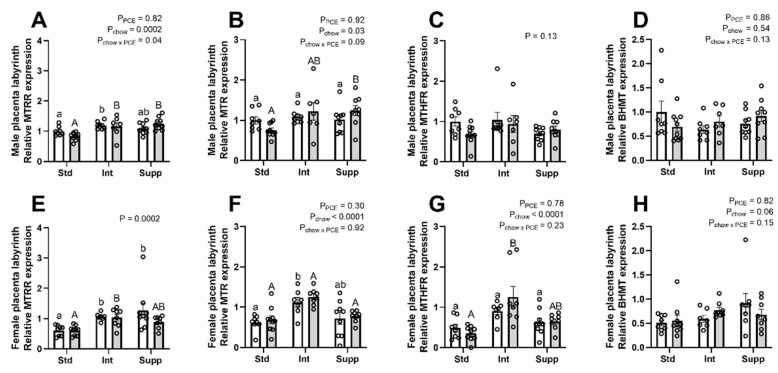
The effects of periconceptional ethanol and maternal choline supplementation on expression of key enzymes of one-carbon metabolism in placentas of male (**A**–**D**) and female (**E**–**H**) fetuses. *Mtrr* (methionine synthase reductase); (**B**,**F**) *Mtr* (methionine synthase); (**C**,**G**) *Mthfr* (5–10 methylenetetrahydrofolate reductase); (**D**,**H**) *Bhmt* (betaine homocysteine methyl transferase). During the periconceptional period, dams received a control liquid diet (Con; open bars) or a liquid diet containing 12.5% *v*/*v* ethanol (PCE; grey bars). For the remainder of pregnancy, dams received either standard chow (Std), intermediate chow (Int), or supplemented chow (Supp), with increasing levels of choline across these groups (1.6 g, 2.6 g, or 7.2 g choline/kg chow), *n* = 7–10 per group. Gene expression was analyzed relative to the geometric mean of *Actb* and *RPL19*, with fold-change relative to the male Std-Con group. All data are presented as mean ± SEM. Significant differences due to alcohol exposure or chow group were identified via two-way ANOVA and Tukey’s post hoc analysis for parametric data, and via the Kruskal–Wallis test across all 6 groups, and Dunn’s post hoc analysis for non-parametric data. Results of post hoc analysis of chow effects are shown by lower case letters for control groups; upper-case letters for alcohol exposure groups.

**Figure 6 nutrients-14-01874-f006:**
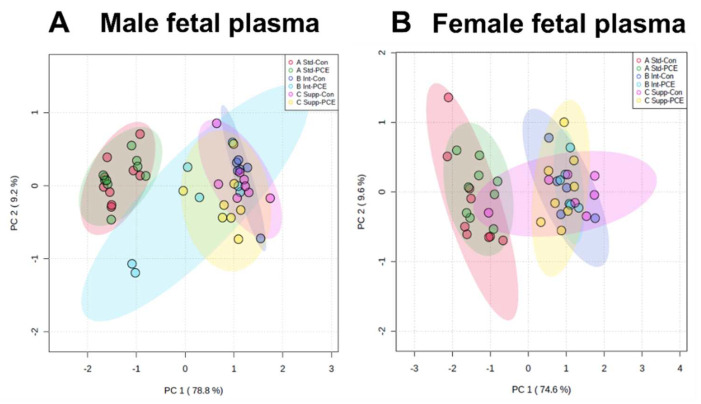
The effects of periconceptional ethanol and choline supplementation on the one-carbon metabolism (1CM) pathway in late-gestation fetal plasma. Principal components analysis (PCA) of 10 molecules from the 1CM pathway across the 6 experimental groups in fetal plasma of (**A**) males and (**B**) females. Molecules were measured using mass spectrometry. Standard chow (1.6 choline/kg chow, Std) with control (0% EtOH, Con) liquid diet (Std-Con, red); Std with periconceptional ethanol (12.5% *v*/*v* EtOH, PCE) liquid diet (Std-PCE, green); intermediate chow (2.6 g choline/kg chow, Int) with Con (Int-Con, dark blue); Int with PCE (Int-PCE, light blue); supplemented chow (7.2 g choline/kg chow, Supp) with Con (Supp-Con, pink); Supp with PCE (Supp-PCE, yellow).

**Figure 7 nutrients-14-01874-f007:**
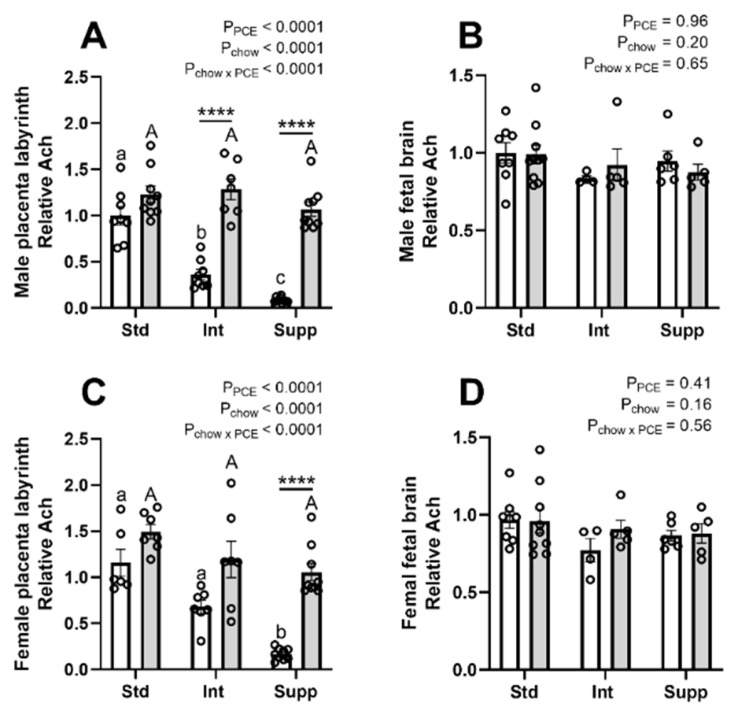
The effects of periconceptional ethanol and choline supplementation on acetylcholine (Ach) levels in placentas of male and female fetuses (**A**,**C**) and male and female fetal brain (**B**,**D**). During the periconceptional period, dams received a control liquid diet (Con; open bars) or a liquid diet containing 12.5% *v*/*v* ethanol (PCE; grey bars). For the remainder of pregnancy, dams received either standard chow (Std), intermediate chow (Int), or supplemented chow (Supp), with increasing levels of choline across these groups (1.6 g, 2.6 g, or 7.2 g choline/kg chow), *n* = 6–9 per group. Molecules were measured using mass spectrometry and data expressed as fold-change relative to the average of the male Std-Con group (mean ± SEM). Significant differences due to alcohol exposure or chow group were identified via two-way ANOVA and Tukey’s post hoc analysis. **** *p* < 0.0001 for differences between Con and PCE within each chow group; chow effects between Con groups are shown by lower-case letters, and between PCE groups by upper-case letters.

**Figure 8 nutrients-14-01874-f008:**
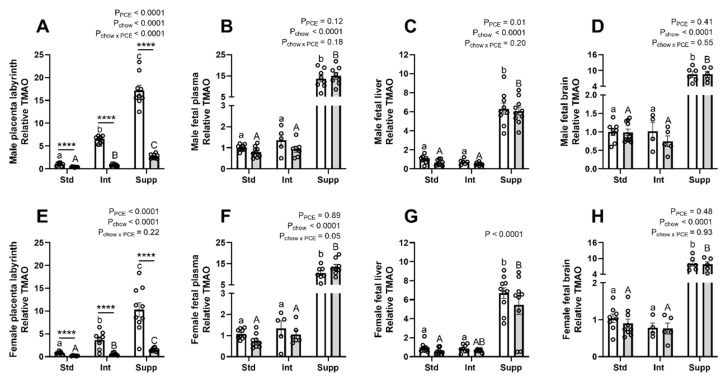
The effects of periconceptional ethanol and choline supplementation on trimethylamine oxide (TMAO) levels in placentas of male and female fetuses (**A**,**E**), male and female fetal plasma (**B**,**F**), male and female fetal liver (**C**,**G**), and male and female fetal brain (**D**,**H**). During the periconceptional period, dams received a control liquid diet (Con; open bars) or a liquid diet containing 12.5% *v*/*v* ethanol (PCE; grey bars). For the remainder of pregnancy, dams received either standard chow (Std), intermediate chow (Int), or supplemented chow (Supp), with increasing levels of choline across these groups (1.6 g, 2.6 g, or 7.2 g choline/kg chow), *n* = 4–9 per group. Molecules were measured using mass spectrometry and data expressed as fold-change relative to the average of the male Std-Con group (mean ± SEM). Significant differences due to alcohol exposure or chow group were identified via two-way ANOVA and Tukey’s post hoc analysis. **** *p* < 0.0001 for differences between Con and PCE within each chow group; chow effects between Con groups are shown by lower-case letters, and between PCE groups by upper-case letters.

## Data Availability

All the processed data are available as manuscript figures or Appendix A.

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
