# Peer review of "Prenatal Choline Supplementation Alters One Carbon Metabolites in a Rat Model of Periconceptional Alcohol Exposure"

_nutrients, 2022, doi:10.3390/nu14091874_

Round 1

Reviewer 1 Report

The original work presented by the authors aimed to systematically evaluate the effect of periconceptional (4d before/ 4d after) alcohol consumption (12.5% vs. 0%; liquid diet) with/without choline post-supplementation (1.6, 2.6 and 7.2 g.100g-1 chow), in one-carbon (IC; folate/methionine/choline) several metabolic intermediaries and related enzymes (at transcription) players in dams´ & fetal´s (plasma/tissues) and placental compartments. The study´s uniqueness lies in the metabolic tracking of fluctuating levels of key IC metabolic intermediates, as compared to other studies on the same subject (doi: 10.1007/s12035-011-8165-5, 10.1146/annurev-animal-020518- 115206) including the authors´(references 12,17).

The study´s experimental design and execution (ethics approval, SBMS/467/14/NHMRC) were impeccable, the analytical variations (errors) seem to be tightly controlled, and the analytical platforms used (LC-MS/MS, qPCR) were quite well selected to pursue the initial hypotheses and objectives. However, due to the many factors (alcohol, choline, tissues, experimental unit) and response variables, is very difficult to depict the overall metabolic impact and dose-response effects (pharmacodynamics), so the authors decide to explain findings by pieces of information. Some modifications are suggested to improve the scientific soundness and uniqueness of the manuscript: 

Title. Too long and suggestive of dose-dependent metabolic integration. Recommendation: Peri-conceptional choline supplementation normalizes the 1C-metabolism of the murine dam/fetus binomial.   

Abstract. Reduce introductory statements and increase the most relevant results in a quantitative>qualitative manner (including p-values). If any, please include the main choline dose-response for key metabolic players.

Introduction. It should be shorter and contextualized to the clinical and epidemiological aspects of alcohol exposure during pregnancy and associated pharmacological treatment, including choline supplementation during pregnancy and peri-conception:

  • Authors can support this the suggested piece of information with references such as (doi): 1136/bmjopen-2016-015410 , 10.1097/AOG.0b013e3182a6b226, 10.1016/S2214-109X(17)30021-9 , 10.3390/nu12061731 , 10.1111/acer.13817 , 10.3390/nu14030688.
  • It is suggested to relocate what is described between lines 44 and 84 (and figure 1) and move it to the discussion section and only leave the background of the authors (previous studies) to outline "the new hypothesis" to be tested in this new manuscript. .

Materials & Methods. OK 

Results. Although surely within the area of perinatology the term "male placenta" is customary, for the general audience it is not, so I suggest using a more appropriate term when needed (e.g., line 387, Figure 7A).

Figures. The resolution of all figures should be improved according to Nutrients´ guidelines. Another way of pointing out the significant differences between treatments should be selected because they are very confusing (lines, asterisks, letters).

Discussion. The authors could use the paragraphs removed from the introduction section (current lines 44-84 + Figure 1) to construct an opening discussion on 1C metabolism, its checkpoints, the effect of exogenous choline (see isotopic studies), and the inhibitory effect of alcohol on this metabolic pathway. Also, maternal-to-placental-to-fetus IC metabolism flux (or cross-talking) should be explained prior to discussing any experimental findings.

  • Regarding the result-by-result discussion, while recognizing the high complexity and quantity of results, a more integrative discussion is needed, if possible, supported by images on the role of choline/alcohol in IC metabolism (eg, see figure 2 in https://www.ncbi.nlm.nih. gov/pmc/articles/PMC3860423/pdf/arcr-35-1-25.pdf).

References. It is suggested to decrease the number of references older than 10 years (currently 34%) to say 25%.

Reviewer 2 Report

In this manuscript, Steane et al examined if prenatal alcohol exposure and choline supplementation affect one-carbon metabolism of fetus and placenta and showed the effects of both alcohol exposure and choline supplementation on placental components of one-carbon metabolism, while the maternal and fetal one-carbon metabolism was predominantly affected by choline supplementation. The topic of this study is important, however, concerns were raised about the descriptive nature of the paper. They examined only levels of components related to one-carbon metabolism and mRNA expression levels of key enzymes such as BHMT and MTHFR. As the authors mentioned, previous report already showed that the exposure to alcohol during this critical time can disturb DNA methylation and gene expression in the early embryo, resulting in an impact on their growth and development. In this study, these authors should explore placental or fetal genes for which the expression changed in response to the alternation of one-carbon metabolism or should show a change in the DNA methylation status by alcohol exposure and choline supplementation. They mentioned about TMAO level which is affected n prenatal alcohol exposure, however, the data were not show in Fig 7.

Round 2

Reviewer 1 Report

The authors have modified their manuscript as requested, so it can be accepted in its current format, after reviewing some structural aspects (e.g. figure legends below each figure, as close to the text where they are cited).

Author Response

We thank Reviewer 1 for their feedback.  All Figures have now been moved as close as possible to first mention in text. Figure legends have been positioned directly below all figures.

Reviewer 2 Report

This manuscript was improved well according to our comments.

Author Response

We would like to thank Reviewer 2 for their feedback.